# Endoscopic-Assisted Microsurgical Meningioma Resection in the Skull Base via Minicraniotomy: Is There a Difference in Radicality and Outcome between Anterior Skull Base and Posterior Fossa?

**DOI:** 10.3390/cancers16071391

**Published:** 2024-03-31

**Authors:** Thomas Kanczok, Gerrit Fischer, Sebastian Senger, Stefan Linsler

**Affiliations:** 1Department of Neurosurgery, Faculty of Medicine, Saarland University Medical Center and Saarland University, 66421 Homburg, Germany; 2Department of Neurosurgery, Klinikum Bayreuth and Medizincampus Oberfranken FAU, 95445 Bayreuth, Germany

**Keywords:** meningioma, endoscopic assisted, skull base, radicality, neuroendoscopy

## Abstract

**Simple Summary:**

In this study, the authors compare the outcomes of skull base meningioma resection using an endoscopic-assisted microsurgical keyhole approach. Between 2013 and 2019, 71 out of 89 patients underwent this procedure. The analysis included 42 anterior skull base and 29 posterior fossa meningiomas. While both cohorts achieved similar gross total resection rates (80% anterior, 82% posterior), complication rates were higher in the posterior fossa (31%) compared with the anterior skull base (16%). Endoscopic assistance was utilized in 79% of cases, with higher rates of tumor detection in the posterior fossa (58.6%) versus the anterior skull base (33%). Statistical analysis showed significantly greater benefits from endoscopy in the posterior fossa. Overall, endoscopic assistance enhances radicality and provides better anatomical visualization, particularly in the posterior fossa, improving outcomes in skull base meningioma surgery.

**Abstract:**

**Background**: Keyhole-based approaches are being explored for skull base tumor surgery; aiming for reduced complications while maintaining resection success rates. This study evaluates skull base meningiomas resected using an endoscopic-assisted microsurgical keyhole approach, comparing outcomes with standard procedures. **Methods**: Between 2013 and 2019; 71 out of 89 patients were treated using an endoscopic-assisted microsurgical procedure. A total of 42 meningiomas were localized at the anterior skull base and 29 in the posterior fossa. The surgical techniques and use of an endoscope were analyzed and compared in terms of complications, surgical radicality, outcome, and recurrences in the patients’ follow-up. **Results**: The two different cohorts yielded similar rates of GTR (anterior skull base: 80% versus posterior fossa: 82%). The complication rate was 31% for the posterior fossa and 16% for the anterior skull base. An endoscope was used in 79% of all cases. Tumor remnants were detected by means of endoscopic visualization in 58.6% of posterior fossa and 33% of anterior skull base meningiomas. The statistical analysis revealed significantly higher benefits from endoscope use in the posterior fossa cohort (*p* < 0.05). **Conclusions**: The results revealed that endoscopy was beneficial in both locations. The identification of remnant tumor tissue and the benefit of endoscopy were clearly higher in the posterior fossa. Endoscopic assistance is a very helpful tool for increasing radicality, providing a better anatomical overview during surgery, and better identifying remnant tumor tissue in skull base meningioma surgery.

## 1. Introduction

Meningiomas are predominantly benign tumors that develop from the meninges, the protective membranes surrounding the brain and spinal cord. They typically exhibit a slow growth pattern. Approximately 30% of brain tumors diagnosed in adults are meningiomas [1]. There is a wide variety of symptoms, e.g., seizures, paralysis, headache, increased intracranial pressure, or visual impairment, depending on the size and localization of the tumor [2]. These tumors can arise in different regions of the skull, with a peak frequency of 25% parasagittal and falcine. About 30% of diagnosed skull base meningiomas are located in the posterior fossa and 60% in the anterior fossa [3].

If the removal of the tumor is required, a transcranial microsurgical approach is the normal surgical approach used and has been improved over the past decades [4,5]. However, there are limitations to the transcranial approach. Additionally, skull base meningiomas often spread extensively, involving vessels and cranial nerves. Their location behind bony structures, nerves, and vessels poses challenges for visualization using traditional microscopes, often necessitating brain retraction or drilling for optimal access [6]. Therefore, new techniques such as endoscopic guidance were established, leading to a “look around the corner” and a less invasive surgical procedure [7]. The drawback of these more advanced surgical procedures is the need for the endoscopic system itself and the steep learning curve to perform neuroendoscopic procedures.

Most studies regarding neuroendoscopy compare a pure endoscopic approach with a microsurgical approach in lesions of the skull base. The most common conclusion is to treat small lesions via endoscopy, whereas bigger lesions should be targeted using a transcranial approach [8,9]. To our knowledge, there are currently no studies that compare the usage of neuroendoscopy in meningiomas of the anterior cranial fossa vs. posterior fossa in correlation.

The aim of this study was to compare the use and assistance of an endoscope in meningioma in anterior and posterior cranial fossa surgery via keyhole approaches and to point out the differences between the two skull base localizations.

## 2. Materials and Methods

### 2.1. Study Design and Patient Population

This retrospective study was authorized by the ethical committee of the medical association of Saarland (44/21). An acquired database of surgically treated meningiomas localized in the anterior and posterior fossa was reviewed. The surgical procedures were performed by senior neurosurgeons (GF, AS, and SL) at the Department of Neurosurgery, Saarland University, between January 2013 and December 2019. All patients were prospectively followed up until December 2022. 

The inclusion criteria encompassed several key factors: a comprehensive set of preoperative and postoperative images, the confirmation of diagnosis through histo-pathological analysis (indicating meningioma), the availability of a complete patient record with detailed postoperative follow-up, and a recorded video of the surgical procedure. The information collected included medical records, image studies, clinical visits, tumor characteristics, intraoperative and postoperative complications, and clinical outcomes.

During the observation period, 372 patients underwent surgery for meningioma. Therein, 89 meningiomas were localized at the anterior skull base or posterior fossa and complied with the inclusion criteria. In 71 out of the 89 cases of skull base meningiomas, an endoscopic-assisted microsurgical technique was applied. Of these 71 patients, 42 presented with a meningioma localized in the anterior fossa and 29 with a meningioma in the posterior fossa (see Figure 1).

### 2.2. Imaging Studies and Tumor Volume Analysis

All patients in the study underwent both pre- and postoperative magnetic resonance imaging (MRI) with gadolinium as the contrast agent, along with a cranial computer tomography (CT) scan. MRI studies were conducted every 6 months during follow-up. Tumor volume was assessed by measuring the longest lateral extent (a) and rostral-to-occipital extension (b) on the axial MRI plane. This was supplemented by the longest axial diameter (c) from the gadolinium-enhanced coronal T1-weighted images. The volume was calculated as V = 3/4 × pi × a × b × c.

### 2.3. Localization, Surgical Approach, and Size of Approach

The surgical approach depended on the localization of the tumor. The different selected surgical approaches were as follows: subtemporal, frontolateral, retrosigmoidal, supraorbital, infratentorial supracerebellar, and suboccipital. The selection of the surgical approach was based on the preference of the performing neurosurgeon, taking into account factors such as tumor size, localization, and invasion into neighboring structures specific to each individual case. Neuronavigation was used routinely. The size of craniotomy was calculated using the longest extent in axial and coronal CT imaging (postoperatively).

### 2.4. Application of the Endoscope

An endoscopic device was applied in 71 (79%) out of the 89 skull base meningiomas (see Figure 1 for details). In 18 cases, the neurosurgeon performed the complete procedure without an endoscopic-assisted inspection because of a large approach and good overview or because there was no intention of radical surgery from the beginning onward. These cases were excluded from the final analysis in this study.

The used devices were a rigid HD rod lens endoscope (Karl Storz SE, Tuttlingen, Germany) with an outer diameter of 4.0 mm and angles of 0°, 30°, and 70°. The endoscope was maneuvered by the neurosurgeon during the procedure. Scopes with 30° and 70° angled lenses were utilized for tumor inspection and resection and were particularly beneficial for obtaining views around corners. The neurosurgeon inserted the endoscope into the surgical field, and an endoscope-holding arm was available in every case for potential bimanual resection under pure endoscopic view. The utility of the endoscope was assessed by the neurosurgeon, focusing on its ability to provide helpful additional information during surgery. This included detecting remnant tumor tissue or neurovascular conflicts that were not visible with the microscope, ensuring complete tumor resection, and enabling the removal of residual tumor tissue under endoscopic view, particularly in challenging areas. Additionally, the endoscope provided valuable information about the anatomical relationships of the tumor with cranial nerves, brainstem, and vessels. Thereby, the analysis was especially focused on the helpfulness of the endoscope, maneuverability, additional information, and changes in intraoperative strategies or related intraoperative complications. All endoscopic- and microscopic-performed steps in the procedures were video recorded. The performing neurosurgeon was interviewed after surgery about handling and satisfaction using the endoscope, with a standardized questionnaire.

### 2.5. Evaluation of Tumor Extirpation

Gross total resection (GTR) refers to complete tumor removal with resection or coagulation of the tumor origin at the dura. Near total resection (NTR) was characterized by >90% tumor removal, while subtotal resection (STR) involved <90% tumor removal. Surgical radicality was assessed using pre- and postoperative MRI imaging.

### 2.6. Clinical Outcome and Follow-Up

Complications and surgical outcomes were analyzed using data from various sources, including the surgical report, video recordings, medical reports, postoperative imaging, and the follow-up data of the patients. The follow-up investigations were conducted regularly at the neurosurgical outpatient department of Saarland University every 6 months or in response to the emergence of neurological symptoms. Recurrence was defined as new evidence of tumor detection in MR imaging studies, evaluated by an experienced neuroradiologist following previous documented gross total resection (GTR).

### 2.7. Statistics

The analysis and visualization of the data were conducted using IBM^®^ SPSS^®^ Statistics Version 26 (SPSS Inc., Chicago, IL, USA). The patient cohorts were compared using the Whitney U test, Fisher’s exact test, and multivariate analysis (Pearson correlation) to assess the differences between values in the groups. Multivariate analysis was performed based on all data by looking at all possible independent variables and their relationships with one another. Special focus was given to the identification of remnant tumor tissue, contact with neighboring vessels or cranial nerves, the localization of the tumor, and tumor volume and radicality, as well as the complication rate. The significance level was set at *p* < 0.05.

## 3. Results

### 3.1. General Results

Overall, 71 patients with meningiomas who underwent an endoscopic-assisted procedure met the inclusion criteria, out of the total of 89 skull base meningioma procedures (79%). The removal of meningioma in the anterior fossa was performed on 42 patients, and the removal of meningioma in the posterior fossa was performed on 29 patients (see Figure 1 for details). The patient population showed an average age of 59.1 years [SD ± 12.9 years]. There were 56 female and 15 male patients. The overall mean surgical time was 162 min [SD ± 72.3 min]. The mean surgical time for meningiomas of the anterior fossa was 149.9 min [SD ± 70.8 min], and for meningiomas of the posterior fossa was 180.7 min [SD ± 75.8 min]. The mean follow-up time was 53.5 months [SD ± 23.1 months]. A total of 61 meningiomas were graded WHO I° and 10 WHO II° (further details can be seen in Table 1).

### 3.2. Preoperative Surgical Decision for the Used Approach and Size of Approach

Throughout the surgery of all patients in this study, the necessary surgical equipment and expertise for conducting an endoscopic-assisted tumor resection were available. The choice of surgical approach was determined by the surgeon’s experience, the specific characteristics of the meningioma, and the patient. The factors considered in the decision-making process included tumor size, location, preoperative neurological deficits, and prominent anatomical structures. Obtaining the gross total resections was the primary goal in all these analyzed cases. The meningiomas were localized at the sellar region in 8 cases, olfactorius groove in 12 cases, lateral frontobasal region in 4 cases, sphenoid wing in 18 cases, and the cerebellopontine angle in 10 cases, as well as tentorium meningiomas in 6 cases and petroclival meningiomas in 13 cases.

The surgical approaches used were as follows: subtemporal (6; 8.4%), frontolateral (10; 13.9%), retrosigmoidal (21; 29.5%) supraorbital (30; 41.9%), infratentorial supracerebellar (2; 2.9%), and parieto-occipital or suboccipital (2; 2.9%). Further details are illustrated in Figure 2. The size of craniotomy was 5.76 ± 2.43 cm^2^ for anterior skull base approaches and 6.98 ± 1.97 cm^2^ for posterior fossa tumors.

### 3.3. Tumor Characteristics and Resection and Recurrence Rate

The overall average tumor volume of patients included in this study was 22.77 cm^3^ (SD ± 36.85 cm^3^). Meningiomas of the posterior fossa showed an average tumor volume of 21.97 ± 11.77 cm^3^, whereas the tumor volume in the anterior fossa showed a mean value of 30.64 ± 25.55 cm^3^. There was no statistically significant difference in the tumor volume between meningiomas in the anterior or posterior fossa. Seven patients had had meningioma surgery previously and presented with a recurrence or progression of remnant tumor tissue. The complete removal of the tumor (GTR) was achieved in 62 (87.3%) cases, whereas incomplete removal (NTR) was seen in 9 (12.7%) cases. In 38 (90%) cases of meningiomas in the anterior fossa, GTR was achieved, whereas four (10%) patients showed NTR. In the posterior fossa, GTR was proven in 24 (82%) cases, and NTR was seen in 5 (18%) cases. There was no significant difference in radicality between the two cohorts (*p* = 0.32). The details are shown in Table 1.

There were three cases of tumor recurrence: one in the anterior fossa (7.1%) and two in the posterior fossa (6.8%). Three patients underwent reoperation with further resection. One patient with a WHO II meningioma had additional radiation therapy. All tumor remnants in the other patients with NTR were stable without progression in MR imaging during the follow-up time period.

### 3.4. Neurological Impairments and Outcome

A restricted field of view was found preoperatively in two patients with meningiomas at the anterior fossa and in one patient with a tumor in the posterior fossa. Double vision was found preoperatively in two cases in the posterior fossa cohort. Other visual disturbances were found in eight other patients. In two cases of meningioma of the anterior fossa, the visual impairment was better after surgery. The visual impairments of patients with meningioma of the posterior fossa remained unchanged after surgery.

Vertigo was reported by six posterior fossa patients and two patients in the anterior fossa cohort, whereas two patients in the anterior and one patient in the posterior fossa cohort suffered from headaches. The patients with vertigo and headaches reported an improvement in the symptoms after surgery.

Hypacusis or anacusis symptoms presented in 10 cases of meningiomas localized in the cerebellopontine angle (*n* = 7) and petroclival region (*n* = 3). Improvements were achieved in three cases after surgery. Two patients complained of worsening hearing function after surgery. One patient was treated with a cochlea implant during follow-up. In five patients, the preoperative hypacusis remained unchanged after surgery.

### 3.5. Perioperative Complications

Overall, the complication rate revealed significant differences between the two cohorts, with 16% for meningiomas of the anterior skull base and 31% for meningiomas of the posterior fossa (*p* < 0.05). The complications were not related to the use of the endoscope or the surgical technique and were mainly based on the localization of the meningioma and the approach used (see also Figure 3). In detail, one patient from the anterior fossa cohort suffered from an ischemic stroke after surgery. Furthermore, one patient developed an abducens paresis, and one patient revealed a new hyposmosia. In two cases, new hygromas were discovered and needed surgical treatment. One patient suffered from postoperative bleeding into the sellar region and needed revision surgery. One patient developed a CSF fistula after surgery, which was treated successfully with lumbar drainage for 5 days.

In patients with meningiomas of the posterior fossa, the new postoperative neurological impairments were as follows: one case of facial palsy, one case of trochlear paresis, and one patient with new gait ataxia. Two patients suffered from reduced hearing after surgery; one of these patients was treated with a cochlear implant. There were two cases of postoperative hydrocephalus, which were treated with a ventriculoperitoneal shunt, and one patient who underwent immediate surgery for CSF fistula had the leakage stopped after surgery. One patient complained about transient dysphagia for 3 months after surgery, which resolved completely. The details are shown in Table 2.

The rate of complications and the neurological outcomes did not correlate with tumor size, gender, the usage of the endoscope to remove remnant tumor tissue, surgical time, or resection extent in the multivariate and correlation analysis.

### 3.6. Application of the Endoscope

With the utilization of the endoscope (featuring an angled optic), remnant tumor tissue was identified in 31 cases, accounting for 44% of the total. Consequently, the surgical strategy was altered (increased mobilization of neighbored structures, the use of angled instruments, and the repositioning of retractors, repositioning of microscope) in 21 cases (29.5%), leading to the removal of remnant tumor tissue under endoscopic view in 17 cases (23.9%).

In meningiomas of the anterior skull base, remnant tumor tissue that could not be seen with the microscope was detected in 14 cases (33%), of which 3 cases were olfactorius meningiomas, 5 cases were a meningioma of the sphenoid wing, 4 tumors were located in the sellar region, and 3 tumors were frontobasal. In five of these cases, invasion of the meningioma into the anterior skull base, ethmoidal cells, and the sphenoid cavity was detected.

Remnant tumor tissue was removed under endoscopic view in 10 (23.8%) cases of meningioma of the anterior skull base. In 32 of these 42 cases (76.2%), the use of an endoscope was considered helpful by the performing surgeon and induced a change in the surgical strategy or improved radicality. Further details are illustrated in Figure 4 and Table 3.

There were no intraoperative complications or technical issues associated with the use of the endoscope: there was no contact with the endoscope to eloquent structures or cranial nerves and no long exposure to the light of the endoscope. Time for endoscopic inspection comprised 75 ± 24 s. The maneuverability and handling of the endoscope were deemed satisfactory by the surgeon in 38 cases (90.4%).

In patients with meningiomas of the posterior fossa, remnant tumor tissue after removal with the microscope was detected in 17 cases (58.6%) via the additional use of an endoscope. Of these 17 remnant tumors, 7 were localized in the petroclival, 7 in the cerebellopontine angle, 2 at the clivus, and 1 at the tentorium. In 10 of the 29 analyzed cases, tumor mass was found in the internal acoustic meatus, which increased the benefit of the endoscope during surgery. In all 29 cases, the use of an endoscope was considered helpful by the performing surgeon and induced a change in the surgical strategy or improved radicality. Further details are illustrated in Figure 3 and Table 3. The resection of remnant tumor tissue was performed under endoscopic view in seven (24.1%) procedures.

There were no intraoperative complications or technical issues associated with the use of the endoscope: there was no contact with the endoscope to eloquent structures or cranial nerves and no long exposure to the light of the endoscope. Time for endoscopic inspection comprised 84 ± 29 s. The maneuverability and handling of the endoscope were deemed satisfactory by the surgeon in 26 cases (89%).

The multivariate and correlation analyses revealed greater benefits from endoscope use in the posterior fossa meningioma cohort than was the case for patients with meningiomas of the anterior skull base (*p* < 0.05). Thereby, the identification of remnant tumors, contact with neighbored vessels or cranial nerves, the localization of tumor, tumor volume, and radicality, as well as the complication rate, were analyzed. The usefulness of the endoscope was based on more detailed information about the tumor and the neighboring structures during surgery and the higher probability of identifying remnant tumor tissue (33% anterior skull base vs. 58.6% posterior fossa). The details are demonstrated in Table 4 and Table 5. Thus, radicality was significantly increased in both cohorts; thereby, remnant tumor tissue was mainly detected in the internal acoustic meatus, intrasellar, and ethmoidal cells.

## 4. Illustrative Cases

### 4.1. Case 1: Endoscopic-Assisted Supraorbital Keyhole Approach

A 68-year-old woman presented with progressive left-sided visual dysfunction. The MR imaging showed a 26 × 30 × 24 mm lesion on the tuberculum sellae with compression of the optic chiasm and extending around the left internal carotid artery. Gross total resection was achieved via a supraorbital keyhole craniotomy. Histopathology confirmed a WHO Grade I meningotheliomatous meningioma. The patient’s visual loss improved within the first postoperative week, and pituitary function remained stable. The follow-up MR imaging at 3 and 24 months postoperatively showed no residual tumor tissue or recurrence (details are shown in Figure 5 and Appendix A).

### 4.2. Case 2: Endoscopic-Assisted Retrosigmoidal Approach

A 74-year-old woman presented with 3 months of persistent trigeminal neuralgia, diplopia, and headache. The MRI scan showed a tumorous lesion (15 × 19 × 16 mm) in the cerebellopontine angle with contact with the tentorium and Meckel’s cave. Gross total resection of the tumor was achieved using an endoscopic-assisted microsurgical retrosigmoidal approach in a semi-sitting positioning. Upon histopathologic examination, a meningotheliomatous meningioma WHO I was found. The patient’s complaints disappeared directly after surgery. She received MR imaging during follow up at 6, 12, and 24 months after surgery, showing no recurrence and no remnant tumor tissue (see Figure 3 and Appendix A).

## 5. Discussion

The resection of skull base meningiomas remains a challenging procedure. Achieving optimal resection rates and long progression-free survival for these patients is paramount. Over the last few decades, advancements in surgical techniques and devices have led to improvements in resection rates and reductions in complication rates [4]. These challenges have spurred nearly a century of debate over the optimal approach for successful removal. Initially, anterior skull base meningiomas were resected via a unilateral subfrontal approach or pterional approach [10,11,12,13]. While these two approaches represent probably a very frequent route (even today), modified skull base approaches like the orbitozygomatic and orbitopterional approaches are also suggested. Additionally, the keyhole concept, involving an eyebrow skin incision and supraorbital minicraniotomy, was introduced for skull base lesions by Perneczky and other colleagues decades ago [6,14,15,16]. For posterior fossa skull base meningiomas, further refinements and approaches are continually evolving [4,6,15,16,17], while transpetrosal approaches have been recommended for petroclival meningiomas [18,19]. However, despite employing extensive approaches, certain aspects of the skull base may not be adequately exposed, potentially leaving residual tumor tissue. Skull base meningiomas are often concealed behind bony corners, nerves, and vessels. Consequently, visualization in a straight line solely via a microscope can be challenging and may necessitate extensive drilling to achieve adequate exposure or the retraction of the brain [6]. Therefore, new techniques, such as endoscopic assistance, were established, leading to a “look around the corner” and a less invasive surgical procedure [7]. Nevertheless, radicality is still the major factor for an excellent long-term prognosis for these patients [4,9]. With growing experience, neurosurgeons can navigate a steep learning curve to tackle more complex pathologies using an array of devices and technical support in skull base surgery of both the anterior and posterior fossa. Numerous authors have already showcased the benefits of employing an endoscopic-assisted microsurgical technique, which aims to achieve greater radicality with reduced surgical trauma and smaller approaches in skull base surgery [6,9,15,20,21,22,23].

However, specialists continue to debate over the optimal surgical technique for enhancing the extent of tumor resection and reducing the complication rate associated with meningiomas of the skull base. Currently, there is no comparison available regarding the surgical technique and the benefits of endoscopic-assisted techniques for both the anterior skull base and the posterior fossa. The authors were able to close this lack of information with this presented series: In total, 71 meningiomas, 42 localized at the anterior fossa and 29 in the posterior fossa, were evaluated for overall outcome and the intraoperative use of an endoscope. Thereby, the endoscope was considered beneficial in all cases of posterior fossa and in 76.2% of anterior fossa meningioma. Both approaches yielded similar results for GTR, NTR, recurrence rate, and postoperative outcome. Overall, GTR was favorable, with 87.3% in this series. The complication rate was significantly higher in the cohort of posterior fossa meningiomas (31% vs. 16%). In this cohort, this was mainly caused by the localization and affection of the cranial nerves in the cerebellopontine angle, as well as brain stem affection. The complication rates of both cohorts were comparable to further published surgical series [4,6,9,20,21,23,24,25,26,27,28,29]. There were no complications related to the handling of the endoscope, nor extensive exposure to the hot lens of the scope, nor direct contact of endoscope and cranial nerves during inspection.

In terms of meningiomas of the anterior skull base, there are several studies comparing the usefulness of an endoscope in removal of tumors of the anterior fossa via an endonasal approach, and the general consensus is that neuroendoscopy can help to identify and remove remnant tumor tissue [6,9,30]. Similar results have previously been reported by the authors as well [9]. Comparisons of data and surgical series using endoscopic-assisted surgery of the posterior fossa in correlation to the anterior and middle fossa are still missing from the literature.

In the presented analysis, the neurosurgeons considered the endoscope to be helpful during the surgery, and it increased radicality in 85.9% of all cases. Additionally, the authors used endoscopy at a much higher frequency for skull base meningiomas than was reported by other authors in previous surgical series [6]. Thereby, the benefit of the endoscope was significantly higher during meningioma surgery in the posterior fossa (100%). This result may be attributed to the anatomical complexities and the chosen approach to the posterior fossa. Achieving an adequate anatomical overview is more challenging in the posterior fossa, and the endoscope provides more detailed information regarding neurovascular conflicts and residual tumor tissue located behind the brainstem and cranial nerves and within the internal acoustic meatus. Even for a keyhole approach to the anterior skull base, the anatomical overview may be comparatively superficial when compared with the retrosigmoid approach. Furthermore, there are more subtle cranial nerves, arterial branches, and perforators localized in the posterior fossa. This fact clearly increases the perioperative complication rate.

Remnant tumor tissue was detected with the endoscope in more than 50% of the cases. Interestingly, the detection of remnant tumor tissues was completely contrary to a report by Schroeder et al., who reported 100% identification of remnant tumor tissue using a supraorbital approach and 56% using a retrosigmoid approach with the endoscope [6]. The authors can only speculate that the supraorbital approach might be used more frequently in that institution and that the neurosurgeons might be more experienced in performing larger exploration via this approach compared with the other group.

The microsurgical resection of skull base meningiomas has been the treatment of choice for many decades. Excellent results have been reported [10,16,17,31,32]. Thus, the question that arises is this: is an endoscopic-assisted technique really necessary for skull base meningiomas? In the authors’ opinion, the endoscopic-assisted technique is very helpful in selected cases of skull base meningiomas, especially if the intention of the neurosurgeon is a keyhole approach and the minimization of surgical trauma in addition to a GTR, in which cases endoscopic assistance should be recommended. In these cases, the endoscope is very helpful and sometimes even mandatory to achieve GTR, as was reported previously [6,9]. Tumor parts in the olfactory groove and intrasellar and invasion into the ethmoidal cells cannot be visualized with the microscope only. Furthermore, the endoscopic view under the ipsilateral optic nerve into the opticocarotid window (behind the internal carotid artery) or the visualization of the diaphragm sellae proves to be invaluable for the neurosurgeon. The authors observed significant benefits from using the endoscope in cases where meningiomas extended into the internal acoustic meatus. Preservation of hearing was a priority, necessitating careful drilling limited to the vestibule and posterior semicircular canal. However, the angle of the microscopic view did not facilitate direct inspection of the internal acoustic meatus fundus. Utilizing a 30° or 70° angled optic provides excellent visualization of the fundus and any residual tumor tissue. This finding is also in line with reports by other colleagues [6,23]. Furthermore, the endoscopic-assisted technique may be used for removing tumor parts in the jugular and suprameatal tubercle, as well as looking into Meckel’s cave. Extensive microsurgical skull base surgeries in these areas might be specifically minimized in the future with angled instruments and a curved high-speed drill under endoscopic view. Although the endoscope was only used for about 2 min (mean) during the whole surgical procedure in the authors’ series, it was truly beneficial for the identification of remnant tumor tissue and providing a better overview of the anatomical structures. The new information visualized via an endoscope can induce a change in surgical strategy in some cases. Additionally, the amount of drilling required for the ideal exposure at the anterior skull base or at the internal acoustic meatus could be reduced, as well as the risk of damaging the cochlea or the destruction of the anterior skull base with a consecutive CSF leakage.

In the authors’ opinion, the endoscope is an essential tool for successfully finishing the surgery when using keyhole approaches for the anterior skull base or posterior fossa. Endoscope-assisted techniques in the field of skull base surgery have proven to be beneficial regarding the following aspects [6,12,33]: the endoscopes offer several advantages in neurosurgery, including increased illumination in the surgical field, enhanced visualization of anatomical structures in close proximity, and the ability to view around corners using angled optics. However, it is essential to recognize that the endoscopic view lacks three-dimensional depth, requiring surgeons to be proficient with specialized devices and anatomical knowledge. Adequate training for hand-eye coordination in the endoscopic view is crucial. Moreover, the use of endoscopes may limit the range of instrument maneuverability based on the approach angle and depth. Despite these considerations, our study demonstrated significant improvements in treatment quality, with endoscopes enabling the detection of more residual tumor tissue and achieving higher radicality without added surgical complications. Areas where the endoscope is likely to be indispensable for resection control include the intrasellar space and the internal acoustic meatus. Comparable results are presented in the previous reports of Marx et al. and Schroeder et al. [6,25].

### 5.1. Summary of the Key Results

The presented study reflects a single-center experience with endoscopic-assisted transcranial minimally invasive skull base surgery for meningiomas. The findings demonstrate a high rate of gross total resection (GTR) and an increase in the radicality of meningioma removal, with complication rates comparable to those in recent publications. The use of an endoscope proved to be particularly beneficial, especially in the posterior fossa, allowing for a higher radicality of resection. The use of an endoscope allowed the avoidance of major optic apparatus and pituitary stalk manipulations in the anterior fossa and to the CN VII, VIII, and V, as well as perforators in the posterior fossa. Furthermore, the extent of drilling of the skull base can clearly be reduced without a reduction in the optimal exposure of the surgical field and the amount of brain retraction. This fact might reduce the perioperative complication rate.

### 5.2. Limitations

The presented study has limitations, including the relatively short-term follow-up and the small numbers of patients in each cohort. Furthermore, the results may be influenced by the personal preferences of the operating neurosurgeon, impacting the choice of approach based on tumor size, localization, and familiarity with endoscopic techniques. These factors could introduce bias into the authors’ findings.

## 6. Conclusions

Despite differences in the tumor localization and surgical approach, we conclude that endoscopic-assisted microsurgical resection of meningiomas was beneficial in both localizations. However, the identification of remnant tumor tissue and the benefit of the endoscope were notably higher in the posterior fossa. Importantly, no significant differences were observed in the extent of resection, postoperative outcomes, or recurrence rates between the two cohorts. The use of neuroendoscopy proves to be a safe and effective procedure for removing meningiomas located in either the anterior or posterior cranial fossa. It serves as a valuable tool for increasing the radicality of resection and reducing the complication rate by providing a better anatomical overview during surgery and aiding in the identification of remnant tumor tissue, particularly in extensive skull base meningioma surgery.

## Figures and Tables

**Figure 1 cancers-16-01391-f001:**
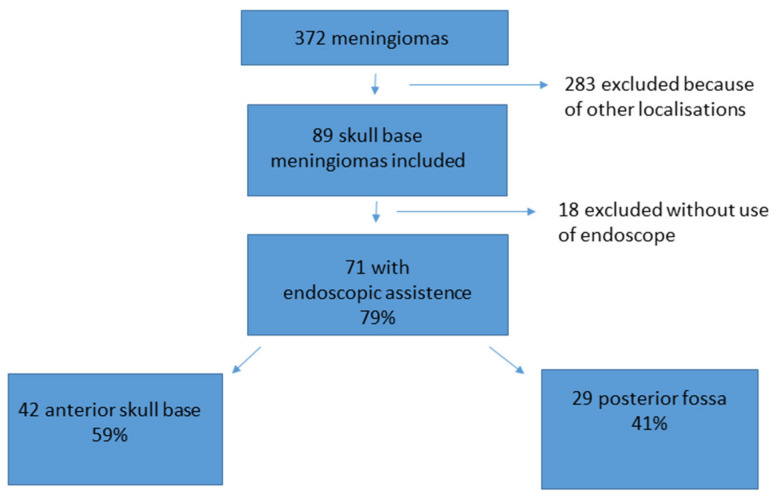
Patient acquisition and data selection. From 372 meningioma patients, 89 met the inclusion criteria of a skull base meningioma. A total of 71 out of the 89 (79%) cases were treated with an endoscopic-assisted microsurgical keyhole procedure. A total of 42 cases were localized at the anterior skull base, and 29 cases were localized in the posterior fossa.

**Figure 2 cancers-16-01391-f002:**
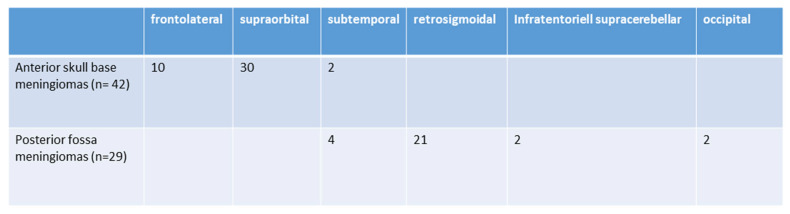
Selected surgical approaches (*n* = 71).

**Figure 3 cancers-16-01391-f003:**
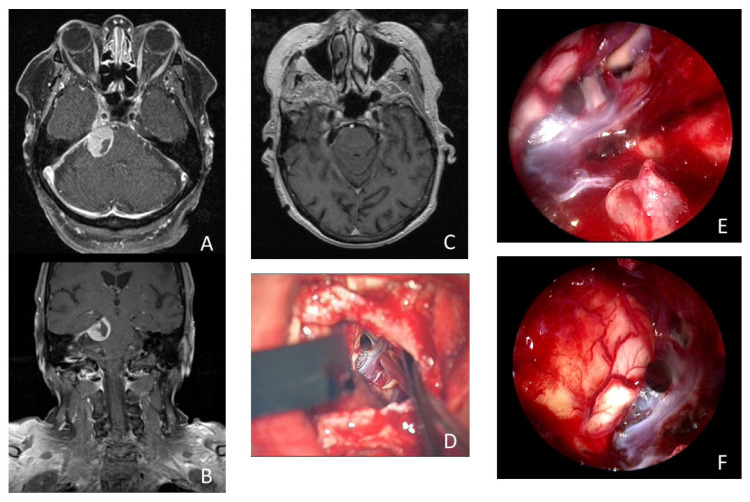
Illustrative case of a 74-year-old woman who presented with 3 months of persistent trigeminal neuralgia, diplopia, and headache. The MRI scan showed a tumorous lesion (15 × 19 × 16 mm) in the cerebellopontine angle with contact with the tentorium and Meckel’s cave in axial (**A**) and coronal (**B**) imaging. Postoperative MRI showed no remnant tumor tissue (**C**). (**D**) Microscopic view of the surgical field after resection of the meningioma. Intraoperative inspection with 30° endoscope revealed remnant tumor tissue (*) in the internal acoustic meatus (**E**). Intraoperative inspection with 30° endoscope into the prepontine space without remnant tumor tissue (**F**).

**Figure 4 cancers-16-01391-f004:**
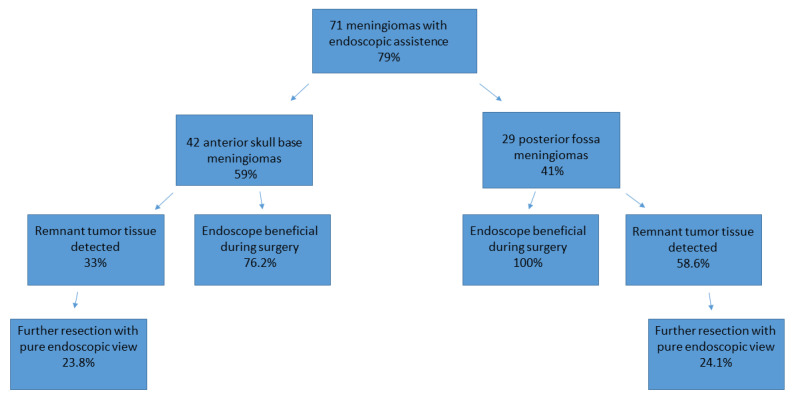
Use of the endoscope and the identification of remnant tumor tissue, illustrated for both cohorts.

**Figure 5 cancers-16-01391-f005:**
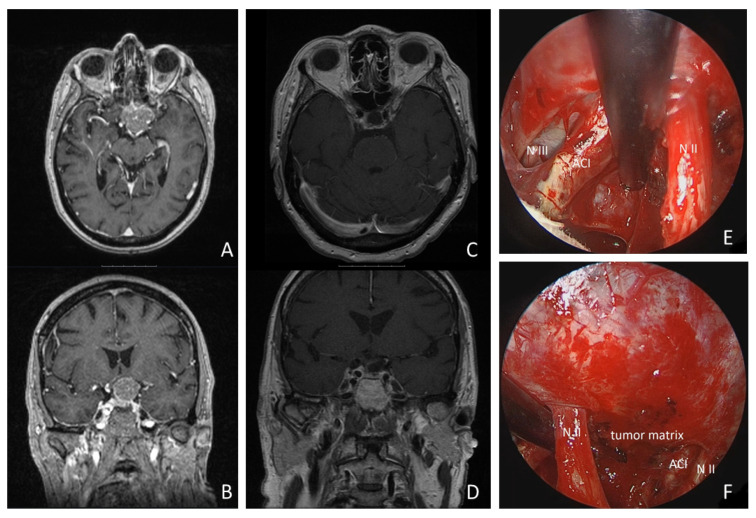
Illustrative case: a 68-year-old woman presented with progressive left-sided visual loss over 1 month. MRI revealed a 26 × 30 × 24 mm tumorous lesion on the tuberculum sellae, compressing the optic chiasm and extending around the left internal carotid artery (see MR imaging axial: (**A**), coronal: (**B**)). Postoperative MRI revealed no residual tumor tissue (see (**C**,**D**)). Gross total resection was achieved using a supraorbital keyhole approach with endoscopic assistance. Intraoperative inspection of the opticocarotid window with a 30° endoscope is shown (see (**E**)). (N II: left optic nerve; ACI: internal carotid artery; N III: N. oculomotorius). * = remnant tumor tissue. (**F**): intraoperative inspection of intrasellar region with 30° endoscope (N II: left optic nerve; ACI: internal carotid artery).

**Table 1 cancers-16-01391-t001:** Correlation of the different meningioma cohorts for posterior fossa and anterior skull base; ns = nonsignificant.

Variables	Meningiomas of the Anterior Skull Base (*n* = 42)	Meningiomas of the Posterior Fossa (*n* = 29)	*p*
Tumor volume	30.64 ± 25.55 cm^3^	21.97 ± 11.77 cm^3^	ns
Histopathology	WHO I: 36 WHO II: 6	WHO I: 25 WHO II: 4	ns
Surgical time	149.9 ± 70.8 min	180.7 ± 75.8 min	ns
Surgical radicality	GTR: 38 NTR: 4	GTR: 24 NTR: 5	ns
Recurrence rate	3 (7.1%)	2 (6.8%)	ns
Complications rate	7 (16%)	9 (31%)	*p* < 0.05
Benefit of endoscope	37 (88%)	29 (100%)	*p* < 0.05
Endoscopic-detected remnant tumor tissue	14 (33%)	17 (58.6%)	*p* < 0.05
Craniotomy size	5.76 ± 2.43 cm^2^	6.98 ± 1.97 cm^2^	ns

**Table 2 cancers-16-01391-t002:** Complications in both patient cohorts.

Variables	Meningiomas of the Anterior Skull Base (*n* = 42)	Meningiomas of the Posterior Fossa (*n* = 29)
CSF fistula	1	1
Ischemic stroke	1	0
CN palsy	1	2
New visual deficits	0	0
New hearing deficits	0	2
Hormonal deficits	0	0
Hygroma	2	0
Postoperative bleeding	1	0
ataxia	0	1
Postoperative hydrocephalus	0	2
Transient dysphagia	0	1

**Table 3 cancers-16-01391-t003:** Use of the endoscope and the identification of remnant tumor tissue via endoscopic view in both cohorts in detail).

Variables	Meningiomas of the Anterior Skull Base (*n* = 42)	Meningiomas of the Posterior Fossa (*n* = 29)
Benefit of endoscope	37 (88%)	29 (100%) *
Satisfaction with handling the endoscope	38 (90.4%)	26 cases (89%)
Endoscopic-detected remnant tumor tissue in - the internal acoustic meatus - cerebellopontine angle - petroclival - intrasellar - ethmoid cells and sphenoid cavity	14 (33%) -- -- -- 5 5	17 (58%) * 7/10 7 7 -- --

* = significant *p* < 0.05.

**Table 4 cancers-16-01391-t004:** Multivariate analysis: impact of variables on complications and neurological outcome.

Variables	aOR (95% CI)	*p* Value
Use of endoscope for resection	1.35 (0.67–1.67)	0.355
Use of endoscope for inspection	1.0 (1.0–1.0)	0.243
Tumor volume	1.25 (0.96–1.45)	0.12
Gender		
Male	Reference	Reference
Female	1.05 (0.96–1.18)	0.288
Surgical time	1.12 (0.85–1.21)	0.211
Radicality	0.96 (0.85–1.10)	0.240
Localisation	0.82 (0.62–1.08)	0.001

**Table 5 cancers-16-01391-t005:** Pearson correlation analysis: impact of variables on radicality and outcomes in correlation to localisation in posterior fossa and anterior skull base.

Variables	Variables	*p* Value
Use of endoscope for inspection	Identification of remnant tumor posterior fossa	<0.01
Use of endoscope for inspection	Identification of remnant tumor anterior skull base	0.04
Tumor volume	Complications	0.03
Tumor volume	Radicality	0.21
contact with neighbored vessels or cranial nerves	Complications	0.01
Localisation	Complications	<0.001

## Data Availability

The raw data supporting the conclusions of this article will be made available by the authors on request.

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
