# Peer review of "Endoscopic-Assisted Microsurgical Meningioma Resection in the Skull Base via Minicraniotomy: Is There a Difference in Radicality and Outcome between Anterior Skull Base and Posterior Fossa?"

_cancers, 2024, doi:10.3390/cancers16071391_

Round 1

Reviewer 1 Report

Comments and Suggestions for Authors

The authors compared the surgical procedures and outcomes of 71 meningioma patients who underwent endoscope-assisted surgery and located at the anterior and posterior skull base.

I believe that the tumor locations and selected surgical approach techniques of the 71 patients are very heterogeneous and it is not possible to standardize their comparison with each other. Also, I do not think it is correct to compare the results of surgical treatment applied in two different anatomy (anterior-posterior fossa).

Comparing the results of surgeries performed with and without endoscope assistance on lesions with the same anatomical location will yield scientifically accurate results. which already includes these in the literature. I do not think that comparing the results of endoscope use in a meningioma located in the anterior skull base with the results of endoscope-assisted surgery in meningiomas located in the posterior fossa will give accurate results. Because the anatomy and surgical approach procedures of the two regions are different. All these conditions cannot be standardized and the use of endoscopes in these two different anatomy cannot be compared with each other.

Also, I have a few suggestions and questions for the manuscript.

1-    Why was the isolated transsphenoidal endoscopic approach not preferred in the 8 hemangioma cases located in the sellar region?

2-    Tumor volume is mentioned in the Results section, but the reader should be clarified about how the tumor volume was calculated in the method section.

3-    It has been mentioned that there are significant differences in terms of complications between tumors located at the anterior and posterior skull base. It should be mentioned what these complications were and what specific complications were significantly different between anterior and posterior? In addition, it was mentioned that complications are not related to the use of endoscope or surgical technique. What data proves this view?

4-    In multivariate analysis, it was mentioned that the complication rate and neurological outcomes were not related to tumor size, gender, endoscope use, surgical time and amount of resection. p values need to be specified and this information needs to be presented to the reader in a table. In addition, in the multivariate analysis, was a statistical study conducted only on complications, or was a statistical study conducted on the resection amounts of anterior and posterior tumors, which is the main theme of the article? If it was done, why wasn't it shared?

5-    In 31 cases, residual tumor was detected with the help of an endoscope, and in 21 of them, it was mentioned that the surgical strategy was changed. But these need to be detailed. Which of the current surgical strategies was switched to?

6-    The fact that the use of an endoscope is considered satisfactory and beneficial by the surgeon is not based on scientific evidence. In order to present this to the reader, the results of cases in which an endoscope was used and those in which an endoscope was not used should be compared with each other and presented in a subjective way.

7-    The Discussion section needs major revision. The vast majority of them contain only literature information. The results of the study should be discussed with the literature.

Author Response

Reviewer 1:

The authors compared the surgical procedures and outcomes of 71 meningioma patients who underwent endoscope-assisted surgery and located at the anterior and posterior skull base.

I believe that the tumor locations and selected surgical approach techniques of the 71 patients are very heterogeneous and it is not possible to standardize their comparison with each other. Also, I do not think it is correct to compare the results of surgical treatment applied in two different anatomy (anterior-posterior fossa).

Comparing the results of surgeries performed with and without endoscope assistance on lesions with the same anatomical location will yield scientifically accurate results. which already includes these in the literature. I do not think that comparing the results of endoscope use in a meningioma located in the anterior skull base with the results of endoscope-assisted surgery in meningiomas located in the posterior fossa will give accurate results. Because the anatomy and surgical approach procedures of the two regions are different. All these conditions cannot be standardized and the use of endoscopes in these two different anatomy cannot be compared with each other.

Thank you for this comment. The idea and concept of the authors was the standarzised analysis of the use of endoscopes in keyhole meningioma surgery and not the correlation of the surgical techiques and outcome in meningioma surgery in the anterior skull base and posterior fossa. The groups and patients are different. The authors agree to that. However, the endoscopic assisted technique is really standarzised and could be evaluated.

Also, I have a few suggestions and questions for the manuscript.

  • Why was the isolated transsphenoidal endoscopic approach not preferred in the 8 hemangioma cases located in the sellar region?

All meningiomas of the sellar region were resected via a supraorbital keyhole approach in this cohort. The decision for this approach in these patients was based on senior author´s experience with both techniques and the preoperative detailed analysis of the individual anatomy of the sellar region and neurovascular encasement in these patients. The authors recommend also the following literature Ref  9. Linsler, S., et al., Endoscopic Assisted Supraorbital Keyhole Approach or Endoscopic Endonasal Approach in Cases of Tuberculum Sellae Meningioma: Which Surgical Route Should Be Favored? World Neurosurg, 2017. 104: p. 601-611.

In conclusion, the surgical approach was chosen based on preference of the performing neurosurgeon considering tumor size, localization and invasiveness into neighboring structures of the individual case. This might be a bias and the authors addressed these aspects in their limitations. See line 467 ff. Furthermore, the results may be influenced by the personal preferences of the operating neurosurgeon, impacting the choice of approach based on tumor size, localization, and familiarity with endoscopic techniques. These factors could introduce bias into the authors´ findings.

2-    Tumor volume is mentioned in the Results section, but the reader should be clarified about how the tumor volume was calculated in the method section.

Thank you for this clue. The formula for the calculation of the volume was added and the method explored in detail. See line 92-97.

3-    It has been mentioned that there are significant differences in terms of complications between tumors located at the anterior and posterior skull base. It should be mentioned what these complications were and what specific complications were significantly different between anterior and posterior? In addition, it was mentioned that complications are not related to the use of endoscope or surgical technique. What data proves this view?

The complications are listed in table 2 in detail (line 249). The major complications in posterior skull base surgery was new hearing deficits, CN palsy and hydrocephalus. These complications were significantly higher than in the cohort of anterior skull base. This is in line with the literature. The complications were not related to the use of endoscopes. This means that there was no contact with the endoscope and cranial nerves directly or a long exposure to the light of the endoscopes. The endoscopic inspection was 75 sec / 84 sec mean. This aspect was added to the revised manuscript. See line 274 ff. and line 288 ff.

4-    In multivariate analysis, it was mentioned that the complication rate and neurological outcomes were not related to tumor size, gender, endoscope use, surgical time and amount of resection. p values need to be specified and this information needs to be presented to the reader in a table. In addition, in the multivariate analysis, was a statistical study conducted only on complications, or was a statistical study conducted on the resection amounts of anterior and posterior tumors, which is the main theme of the article? If it was done, why wasn't it shared?

The authors agree. The informations and data are presented in the new table 4 and table 5 with p values for better understanding. See also informations line 177 ff. and line 323 ff.

5-    In 31 cases, residual tumor was detected with the help of an endoscope, and in 21 of them, it was mentioned that the surgical strategy was changed. But these need to be detailed. Which of the current surgical strategies was switched to?

Thank you for this important comment. The surgeon had to decide how to increase the radicality when remnant tumor tissue was deteceted. This could be achieved by more mobilization of neighboured structures, use of angled instruments or another positioning of the retractors and microscope. The authors added this informations to the manuscript. See line 253 f.

6-    The fact that the use of an endoscope is considered satisfactory and beneficial by the surgeon is not based on scientific evidence. In order to present this to the reader, the results of cases in which an endoscope was used and those in which an endoscope was not used should be compared with each other and presented in a subjective way.

This study excluded the cases without an use of a endoscope completey. The authors agree to reviewers´ comment. However, the intention of this manuscript was not the comparison of cases without and with endoscope. This might be an additional scientific work with different cohorts. The main intention of this manuscript is the comparison of an endoscopic assistend microsurgical technique in the anterior skull base and posterior fossa. Therefore, the objective analysis of the use of an endoscope for meningimoma removal via keyhole approach is not really necessary to adress the aim of this study. The authors interviewed the performing surgeon after surgery about handling and satisfaction using the endoscope. This information have been added tot he revised manuscript as well.

7-    The Discussion section needs major revision. The vast majority of them contain only literature information. The results of the study should be discussed with the literature.

The Discussion section was revised completely and the authors´s results discussed more in correlation to the recent literature. See line 367 ff.

Reviewer 2 Report

Comments and Suggestions for Authors

The Authors present a retrospective analysis of endoscope-assisted surgery for meningiomas located in the anterior and posterior fossa. The methodology is solid, and the aim of this study is well-presented and quite interesting. The use of keyhole approaches with the aid of the endoscope for skull base surgery is a trending topic in neurosurgery and the results presented and discussed by the Authors are in line with data retrieved from current Literature. Overall, the quality of the paper is good, but there are some minor issues to be solved:

- Several typos and grammar mistakes should be corrected.

- Adding graphics of the statistical analysis would improve the presentation of this case series.

Comments on the Quality of English Language

There are several typos and grammar mistakes. Overall, the meaning of the paper is clear, but corrections are obviously required

Author Response

Dear Editors,

We thank the referees for their helpful and constructive comments. Attached are the comments of the referee reports and descriptions of how we addressed them in detail.

Reviewer 2:

The Authors present a retrospective analysis of endoscope-assisted surgery for meningiomas located in the anterior and posterior fossa. The methodology is solid, and the aim of this study is well-presented and quite interesting. The use of keyhole approaches with the aid of the endoscope for skull base surgery is a trending topic in neurosurgery and the results presented and discussed by the Authors are in line with data retrieved from current Literature. Overall, the quality of the paper is good, but there are some minor issues to be solved:

- Several typos and grammar mistakes should be corrected.

The manuscript has been edited completely by the English Language Editing Services of MDPI.

- Adding graphics of the statistical analysis would improve the presentation of this case series.

Thank you for this remark. The authors have added this information in the revised manuscript. See e.g. table 4 and table 5 and revised chapter results and discussion.

Reviewer 3 Report

Comments and Suggestions for Authors

The authors presented their institutional surgical results of endoscopy-assisted microsurgical skull base meningioma resection via minicraniotomy. They compared between anterior and posterior skull base lesions.

Use of an endoscopy enabled to detect tumor remnants more often in the posterior fossa lesions (58.6%, vs anterior skull base 33%). 

They concluded endoscopic assistance is helpfull to increase the radicality with a better anatomical overview and identification of remnant tumor.

Overall, the article is well-written and contains sufficient interest. Some minor points to be addressed.

・line 139

“multivariate analysis to assess differences between in the groups.”

This should be more describe in detail.

・line 234 “In meningiomas of the anterior skull base remnant tumor which could not be seen with the microscope was detected in 14 cases (33%)”

  • I suppose this should be started in a new paragraph for better readability.

・line 234 “In meningiomas of the anterior skull base remnant tumor which could not be seen with the microscope was detected in 14 cases (33%)”

line 239 “Remnant tumor tissue was removed under endoscopic view in 10 (23.8 %) cases of meningioma of the anterior skull base.”

line 253 “In patients with meningiomas of the posterior fossa remnant tumor after removal with the microscope were detected in 17 cases (58.6%) by additional use of an endoscope.”

line 260 “ Resection of remnant tumor tissue was performed under endoscopic view in 7 (24.1 %) procedures.”

  • I think the result of this study is in line with my clinical impressions. There are several cases where endoscopy is useful in detecting residuals, but they cannot always resected in endoscopic view.

・line 265

“The multivariate analysis revealed a higher benefit of the endoscope in the cohort of the posterior fossa meningiomas as in the meningiomas of the anterior skull base (p<0.05).”

  • How multivariate analysis was performed should be described in detail. What variables were included? How variables were selected?

・line 395 “Utilizing a 30° or 70°angled optic provides excellent visualization of the fundus and any residual tumor tissue. This finding is also in line with reports by other colleages[6, 28]. Furthermore, the endoscopic-assisted technique may be used removing tumor parts the jugular and suprameatal tubercle as well as looking into Meckel´s cave. Especially extensive microsurgical skull base surgeries in these areas might be minimized in the future.”

  • looking into these areas may be possible by endoscope, but is it possible to resect the lesion without bone drilling or microsurgical manipulation?

・Perioperative complication rate seems to be high. Is this because of the narrow surgical field with minicraniotomy?

Comments on the Quality of English Language

Throughout the article, many grammatical or typographical errors are detected. I recommend English editing.

Round 2

Reviewer 1 Report

Comments and Suggestions for Authors

After the necessary edits, the study has become more scientific for the reader.

Author Response

Thank your for your review.

The authors reduced the references. Now, there are only cited the really essential references of the authors and co-authors. We think that these references are essential for background and are valid. Therefore, we feel the new self-citation rate as completely adequate.

Additionally, some sentences were rewritten as proposed to reduce overlapping with previous publications. Now, there is no overlap any more to your opinion.

We hope to have addressed the suggestions of the reviewers which have helped in the improvement of this manuscript, and that our profound revised work is now acceptable for publication in the Special Issue of Cancers finally.

Sincerely

Stefan Linsler